# Urea Transporter Inhibitor 25a Reduces Ascites in Cirrhotic Rats

**DOI:** 10.3390/biomedicines11020607

**Published:** 2023-02-17

**Authors:** Yi Ying, Nannan Li, Shuyuan Wang, Hang Zhang, Yinglin Zuo, Yiwen Tang, Panshuang Qiao, Yazhu Quan, Min Li, Baoxue Yang

**Affiliations:** 1Department of Pharmacology, School of Basic Medical Sciences, Peking University, Beijing 100191, China; 2The State Key Laboratory of Anti-Infective Drug Development, Sunshine Lake Pharma Co., Ltd., Dongguan 523871, China; 3Key Laboratory of Molecular Cardiovascular Sciences, Ministry of Education, Beijing 100191, China

**Keywords:** urea transporter inhibitors, diuretics, dimethylnitrosamine, cirrhotic ascites

## Abstract

Ascites is a typical symptom of liver cirrhosis that is caused by a variety of liver diseases. Ascites severely affects the life quality of patients and needs long-term treatment. 25a is a specific urea transporter inhibitor with a diuretic effect that does not disturb the electrolyte balance. In this study, we aimed to determine the therapeutic effect of 25a on ascites with a dimethylnitrosamine (DMN)-induced cirrhotic rat model. It was found that 100 mg/kg of 25a significantly increased the daily urine output by 60% to 97% and reduced the daily abdominal circumference change by 220% to 260% in cirrhotic rats with a water intake limitation. The 25a treatment kept the serum electrolyte levels within normal ranges in cirrhotic rats. The H&E and Masson staining of liver tissue showed that 25a did not change the cirrhotic degree. A serum biochemical examination showed that 25a did not improve the liver function in cirrhotic rats. A Western blot analysis showed that 25a did not change the expression of fibrosis-related marker protein α-SMA, but significantly decreased the expressions of type I collagen in the liver of cirrhotic rats, indicating that 25a did not reverse cirrhosis, but could slow the cirrhotic progression. These data indicated that 25a significantly reduced ascites via diuresis without an electrolyte imbalance in cirrhotic rats. Our study provides a proof of concept that urea transporter inhibitors might be developed as novel diuretics to treat cirrhotic ascites.

## 1. Introduction

Ascites is a typical symptom of liver cirrhosis that is caused by a variety of liver diseases. Ascites occurs in 5~10% of compensated cirrhotic patients each year [1]. Patients with cirrhotic ascites are prone to secondary complications such as bacterial infections, electrolyte disturbances, kidney injuries and malnutrition [2,3]. A few patients with refractory ascites do not respond to treatment or relapse quickly, with an average survival of 6 months [4,5]. Ascites severely affects the life quality of patients and increases the burden on family and society.

At present, the classical diuretics used for cirrhotic ascites mainly include an aldosterone antagonist (spironolactone) and loop diuretics (furosemide, torasemide and bumetanide) [6,7], which increase the urine output following sodium excretion. The long-term use of these drugs may change the levels of blood sodium, potassium and chloride and may cause electrolyte imbalances such as hyponatremia, hypokalemia and hyperkalemia, which can lead to severe secondary effects such as cardiac arrhythmias and sudden death [8]. In recent years, vasopressin V2 receptor antagonists such as tolvaptan as a novel class of diuretics were used clinically to treat diseases with water retention. Tolvaptan has the advantage of increasing the urine output without influencing sodium and potassium excretion. However, tolvaptan has several side effects such as gastrointestinal bleeding [9,10] and liver damage [11]. Therefore, it is necessary to develop novel diuretics that are suitable for long-term use without disturbing the electrolyte balance.

Urea transporters (UTs) are a family of membrane channel proteins that are selectively permeable to urea and mediate urea transport across the plasma membrane. There are at least 4 UT members, including UT-A1, UT-A2, UT-A3 and UT-B [12], that are expressed in the descending limbs, the inner medullary collecting duct and descending vasa recta, respectively, and involved in intrarenal urea recycling. Intrarenal urea recycling plays a crucial role in the urine concentrating mechanism [13]. In previous studies, UT knockout mouse models showed that UT functional deletion caused a urea-selective urine-concentrating defect without affecting the electrolyte balance [14,15,16,17,18], which suggests that UTs are a novel diuretic target and that UT inhibitors may be developed into novel diuretics to treat patients with water retention [19,20]. The advantage of UT inhibitors is that they act as diuretics without affecting the excretion of Na^+^, K^+^, Cl^−^ and other electrolytes [17,18]; the functional inhibition of UTs by a specific inhibitor did not change urea excretion either [21], making them suitable for the long-term treatment of patients.

In our previous study, a UT inhibitor with a diarylamide scaffold named *N*-(4-acetamidophenyl)-5-acetylfuran-2-carboxamide and denoted as 25a showed a specific inhibitory activity on UTs and a potent diuretic effect without an electrolyte imbalance in rats and mice [22]. Furthermore, 25a had outstanding pharmacological characteristics both in vitro and in vivo over other UT inhibitors previously studied in our laboratory [21,22,23,24,25]. 25a was rapidly absorbed after an oral administration and its absolute bioavailability was 20.1%. In vivo, 25a at 50 mg/kg and 100 mg/kg significantly increased the urine output compared with the controls. In vitro, 25a did not show cytotoxicity under 400 mM, as analyzed by a CCK-8 kit [22,26]. These experimental data suggest that 25a might be suitable to be used to treat cirrhotic ascites.

In the present study, the first effort was made to determine the effect of 25a on ascites in dimethylnitrosamine (DMN)-induced cirrhotic rats. The experimental results showed that 25a decreased ascites in cirrhotic rats via diuresis. In this study, we provide proof of concept for UT inhibitors to be developed as a novel class of candidate drugs for treating cirrhotic ascites.

## 2. Materials and Methods

### 2.1. Rat Cirrhosis Model

Male SD rats were purchased from the Animal Center of Peking University Health Science Center. According to the Guide for the Care and Use of Laboratory Animals published by the US National Institutes of Health, the rats were fed under pathogen-free conditions with a 12 h light/dark cycle at 22 ± 3 °C. Animal experiment was proven by the Institutional Animal Care and Use Committee of Peking University Health Science Center (laboratory animal use license No. SYXK(JING)2021-0064; laboratory animal production license No. SCXK(JING)2021-0013).

Based on the description in a previous report [27] with a few modifications, a cirrhotic rat model was established by an injection of N-Nitrosodimethylamine, DMN (TCI, Shanghai, China, D0761). DMN was diluted with saline to a final concentration of 1% [28]. Cirrhosis was induced by an intraperitoneal injection (i.p.) of DMN (10 mg/kg body weight/day) on 3 consecutive days of each week for 4 weeks. DMN administration was stopped once during the 5th week and restarted from the 6th week to the 8th week on 2 consecutive days of each week. After successful modeling, 21 rats with a similar abdominal circumference, body weight and general condition were selected and randomly divided into 3 groups: the model group (n = 7), the 25a group (n = 7) and the tolvaptan group (n = 7).

The rats were placed in a metabolic cage and the body weight and abdominal circumference were measured. Urine samples were collected every 24 h. After the onset of ascites, a 24 h observation period was conducted, followed by the administration of 25a (100 mg/kg), tolvaptan (3 mg/kg) or a vehicle (0.5% poloxamer) to the rats by gavage. The treatment period was 6 days. The rats were given free access to food. The water intake was limited to the previous day’s intake amount plus 10 mL, as previously described [27]. During the treatment, blood samples were collected from the rats every 3 days. At the end of the treatment, liver and kidney samples were collected from the rats.

### 2.2. Histological Evaluation

The liver and kidney tissue blocks were fixed in 4% paraformaldehyde (Innochem, Beijing, China, B46674) and embedded in paraffin (Leica, Wetzlar, Germany, 39601095). Paraffin sections (5 μm) were prepared (Leica, Wetzlar, Germany, RM2235). Hematoxylin and eosin (H&E) and Masson’s trichrome staining were performed for the histological analysis. The histological examinations of the liver and kidney were evaluated by the authors in a double-blind format.

Liver-cell necrosis was graded from 0 to 5 as follows: (0) no necrosis of hepatocytes; (1) mild focal necrosis alone; (2) focal necrosis with scattered mild centrilobular necrosis; (3) confluent hepatic necrosis; (4) submassive hepatic necrosis; and (5) massive hepatic necrosis [29]. The quantification of liver fibrosis was performed using the Knodell and Ishak scoring system [30] as follows: (1) no fibrosis; (2) fibrous portal dilatation; (3) bridging fibrosis (portal–portal or portal–central junction); and (4) cirrhosis.

Kidney tissue injuries were scored according to a published procedure [31] as follows. (i) A glomerular injury was classified into the following grades: (0) none; (1) <25% of glomeruli exhibiting non-specific features of an injury; (2) 25~50% of glomeruli exhibiting non-specific features of an injury; (3) 50~75% of glomeruli exhibiting non-specific features of an injury; and (4) >75% of glomeruli exhibiting non-specific features of an injury. (ii) Tubular necrosis was classified as follows: (0) none; (1) <25% of tubules of the entire renal parenchyma; (2) 25~50% of tubules of the entire renal parenchyma; (3) 50~75% of tubules of the entire renal parenchyma; and (4) >75% of tubules of the entire renal parenchyma. (iii) Inflammatory infiltrates were classified as follows: (0) none; (1) leucocytes confined within the interstitium; and (2) leucocytes infiltrating the interstitium and tubular epithelial cells.

### 2.3. Blood Biochemical Assay

Blood samples were collected from the rats by an orbital venous plexus puncture with an anticoagulation of sodium heparin [22]. The serum samples were tested with an alanine aminotransferase (ALT) kit (Nan-jing Jiancheng Biochemicals Ltd., Nanjing, China, C009-2-1), an aspartate aminotransferase (AST) kit (Nan-jing Jiancheng Biochemicals Ltd., Nanjing, China, C010-2-1), a blood urea nitrogen (BUN) kit (Nan-jing Jiancheng Biochemicals Ltd., Nanjing, China, C013-2-1) and a creatinine kit (Nan-jing Jiancheng Biochemicals Ltd., Nanjing, China, C011-2-1). The serum sodium, potassium and chloride were determined at Peking University Third Hospital.

### 2.4. Western Blot Analysis

The tissue protein samples were extracted using a RIPA lysis buffer (Applygen, Beijing, China, C1053) containing a protease inhibitor cocktail (Roche, South San Francisco, CA, United States, 11873580001). The proteins were separated based on their molecular weight through electrophoresis and then transferred to polyvinylidene difluoride membranes (Amersham Biosciences). After blocking, the membranes were incubated with anti-COL1A2 (Immunoway, YT6135, 1:1000 dilution) or anti-α-SMA (ABclonal, A7248, 1:1000 dilution). The membranes were then washed 3 times and incubated with goat anti-rabbit IgG secondary antibodies. Blots were then developed with an ECL kit (Meilunbio, Dalian, China, MA0186) and visualized with a chemiluminescence detection system (Syngene, GeneGnome XRQ, Cambridge, UK). Quantitation was performed by scanning and analyzing the intensity of the hybridization bands.

### 2.5. Statistical Analysis

All results were analyzed with GraphPad Prism 9 software (GraphPad, San Diego, CA, USA) and were expressed as the mean ± SEM. A one-way ANOVA with Tukey’s HSD post hoc test was performed for the statistical analysis. A *p*-value < 0.05 was considered to be statistically significant.

## 3. Results

### 3.1. 25a Reduces Ascites in Cirrhotic Rats

DMN caused death in 25.9% of rats during the modeling; the 25a or tolvaptan treatments did not affect the rat survival. Figure 1A shows that the daily abdominal circumference increased much faster in cirrhotic rats than in the control rats (*p* < 0.01), which indicated the ascites production in cirrhotic rats. It was found that 25a at 100 mg/kg significantly decreased the abdominal circumference in cirrhotic rats (*p* < 0.01). The positive control tolvaptan at 3 mg/kg also decreased the abdominal circumference in cirrhotic rats (*p* < 0.05). The rat abdominal circumference was found to have a strong correlation with the body weight (Figure 1C). These results suggest that 25a could significantly reduce ascites in cirrhotic rats.

### 3.2. 25a Increases Urine Output and Decreases Urinary Osmolality in Cirrhotic Rats

With a water intake limitation, the daily urine samples of the rats were collected in metabolic cages. The results showed that the urine output in cirrhotic rats was significantly less than the control rats. It was found that 100 mg/kg of 25a increased the daily urine output by 60% to 97% in cirrhotic rats and 3 mg/kg of tolvaptan also had a significant diuretic effect (*p* < 0.01) (Figure 2A). The urinary osmolality was significantly lower in the cirrhotic rats with the 25a or tolvaptan treatments than in the cirrhotic rats with the vehicle treatment (*p* < 0.05) (Figure 2B), which suggested that 25a had a diuretic activity in cirrhotic rats.

### 3.3. 25a Does Not Affect Blood Osmolality and Electrolyte Concentration in Cirrhotic Rats

To determine the effect of 25a on blood osmolality and electrolytes in cirrhotic rats, the serum samples of the rats were collected and analyzed on day 6 with a continuous treatment. As shown in Table 1, the serum osmolality and sodium and potassium levels were significantly lower in the cirrhotic rats than the control rats (*p* < 0.05), which was reversed by the 25a treatment. Compared with the control group, the cirrhotic rats had a significantly higher serum chloride level, which was not affected by 25a. The aforesaid results suggested that 25a could keep the serum electrolyte levels within normal ranges.

### 3.4. 25a Does Not Reverse Cirrhosis but Reduces Ascites during Cirrhotic Progression

The rat liver tissues and blood samples were collected on the day 6 with a continuous treatment. The H&E staining of the liver tissue section showed hepatocyte destruction, necrosis, degeneration and sinusoid loss in the cirrhotic rats (Figure 3A), confirming that the cirrhosis model was successfully established. The Masson staining of the rat liver tissue showed typical cirrhotic characteristics, including an obvious collagen deposition and hepatocyte fibrosis (Figure 3B). However, we did not find a significantly protective effect of the 25a and tolvaptan treatments for 6 days on the cirrhosis. Moreover, the serum ALT (Figure 3C) and AST (Figure 3D) significantly increased by 100% in the cirrhotic rats (*p* < 0.01), indicating a defective liver function, which did not significantly improve in the rats with the 25a or tolvaptan treatments for 6 days. The cirrhotic rats had significantly lower blood albumin levels than the control rats (*p* < 0.01). Lowered blood albumin levels were improved in the 25a- and tolvaptan-treated cirrhotic rats (*p* < 0.01) (Figure 3E). Furthermore, the liver index was decreased in the cirrhotic rats (*p* < 0.01), which was not reversed by 25a or tolvaptan (Figure 3F). In addition, we detected fibrosis-related marker protein type I collagen (Col-I) and α-smooth muscle actin (α-SMA) in the rat livers from the Western blot analysis. The results showed that the higher expression of α-SMA in cirrhotic rats did not decrease with the 25a or tolvaptan treatments. However, 25a significantly reduced the Col-I expression in the cirrhotic rats (*p* < 0.01) (Figure 3G). These experimental results indicated that 25a did not reverse cirrhosis, but could reduce ascites during the cirrhotic progression.

### 3.5. Effects of 25a on Renal Function in Cirrhotic Rats

Compared with the control rats, the cirrhotic rats showed histopathological damage to the kidneys, which included glomerular lesions and tubule dilatation accompanied by significantly increased blood urea nitrogen (BUN) and creatinine (Crea). There was no significantly histological difference between the 25a- or tolvaptan-treated rats and the cirrhotic rats (Figure 4A). There was no significant difference in the kidney indexes between all groups (Figure 3F). However, 25a decreased the BUN and Crea levels on day 6 with a continuous treatment (*p* < 0.05) (Figure 4B).

## 4. Discussion

Diuretics are commonly used for the treatment of ascites caused by liver cirrhosis [32]. Previous studies have found that 20~40% patients with cirrhotic ascites have adverse reactions related to the diuretic treatment [33]. The reasons are that the long-term or high-dose use of classical diuretics can cause a blood electrolyte imbalance [34], which secondarily leads to a series of physiological changes and pathological syndromes such as cardiac arrhythmias [35]. In addition, the novel diuretic vasopressin V2 receptor antagonist tolvaptan has hepatic toxicity [36]. The motivation of this study was to determine if urea transporter inhibitors [21] could be developed to treat ascites with better pharmacological properties and less adverse reactions. DMN-caused liver cirrhotic rats were used as an animal ascites model, in which a specific urea transporter 25a as a tool drug and tolvaptan as a positive control drug were studied.

DMN-caused liver cirrhotic rats are commonly used to evaluate the pharmacological effect of drugs on ascites. In the present study, liver cirrhosis was successfully established in rats with DMN. The cirrhotic rat model was achieved by utilizing 10 mg/kg DMN over a reasonably short duration (3~4 weeks), which was an advantage over other animal models such as the CCl_4_-induced cirrhotic model [37,38]. Our results showed that DMN caused ascites, characterized by a daily increased abdominal circumference and body weight and a decreased urine output. Tolvaptan as a positive control could partly reverse these ascites-related symptoms. These data suggested that the DMN-induced cirrhotic rat model was suitable to evaluate the effect of 25a on ascites.

In our previous study, 25a was demonstrated to be an orally available UT inhibitor with good pharmacological properties [22]. This study confirmed that an intragastric administration of 25a caused an obvious diuretic effect without causing an electrolyte imbalance in cirrhotic rats, which indicated that 25a had fewer adverse effects than the currently used diuretics in terms of the diuretic impact.

Serum ALT and AST are biochemical markers of liver functional injuries. DMN-induced cirrhotic rats showed severely high levels of serum ALT and AST. We found that the 25a treatment did not affect the increased serum ALT and AST in the cirrhotic rats. 25a did not relieve DMN-induced liver morphological abnormalities such as fibrosis, focal necrosis, the hydropic degeneration of hepatocytes or steatosis, which suggested that 25a could not reverse the cirrhotic condition. The decreased serum albumin levels in the DMN-induced cirrhotic rats resulted from hepatic necrosis; 25a could increase the blood albumin level in the cirrhotic rats. We considered that this phenomenon could be due to a hemoconcentration caused by 25a increased free-water excretion. 25a also improved the reduced liver-to-body-weight ratio, which may have been due to the remission of ascites preventing the progression of cirrhosis. A Western blot analysis of the fibrosis-associated proteins showed that 25a did not reverse the increased α-SMA expression in the livers of the cirrhotic rats. These experimental results indicate that 25a did not improve cirrhosis, but could reduce ascites during the cirrhotic progression.

Hepatorenal syndrome is a type of kidney function injury that typically occurs in association with cirrhosis and raises a therapeutic challenge for patients with advanced liver disease [39]. Thus, we studied the effect of 25a on the kidney tissue morphology and function. Our data showed that 25a did not lead to the aggravation of a kidney injury caused by cirrhosis. The BUN level in the 25a-treated cirrhotic rats was much lower than in the cirrhotic rats without or with a tolvaptan treatment, which might have been due to 25a inhibiting UT-A-mediated urea reabsorption at the end of the renal collecting ducts and reducing the ascites. The 25a treatment could also decrease the level of blood creatinine, suggesting that 25a did not impair the renal function.

However, there were a few limitations to this study. First, the model employed in this work was a DMN-induced cirrhotic rat model, which has a high mortality rate. The major challenge to determine the exact period required to develop cirrhotic ascites in order to avoid the reversibility of a spontaneous liver injury. Second, it is difficult to simulate all the pathological manifestations of ascites with this model. Third, the efficacy of 25a for 6 days in this study was not a long-term treatment. We intend to extend the period of treatment in a subsequent study to investigate the time-response relationship of 25a on ascites.

In conclusion, we discovered the therapeutic effect of urea transporter inhibitor 25a on ascites in a DMN-induced cirrhotic rat model. The pharmacological mechanism of 25a on reducing ascites depends on its diuretic effect. This study further confirmed that 25a demonstrated diuresis without causing a blood electrolyte imbalance, which suggests that UTs are a suitable diuretic target for reducing ascites. UT inhibitor 25a or its derivatives might be developed as novel diuretic drugs for cirrhotic ascites.

## Figures and Tables

**Figure 1 biomedicines-11-00607-f001:**
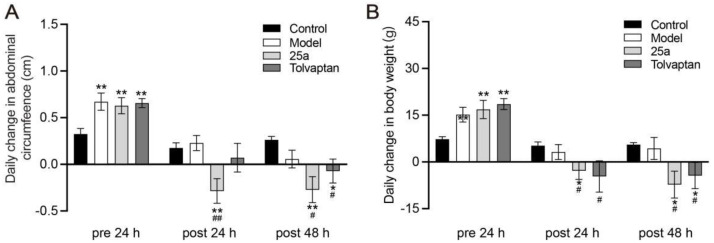
Daily abdominal circumference change and daily body weight change in rats. (**A**) Daily change in abdominal circumference. (**B**) Daily change in body weight. Data are presented as means ± SEM, n = 7. * *p* < 0.05 and ** *p* < 0.01 vs. control group; # *p* < 0.05 and ## *p* < 0.01 vs. model group. The data were analyzed by one-way ANOVA with Tukey’s HSD post hoc test.

**Figure 2 biomedicines-11-00607-f002:**
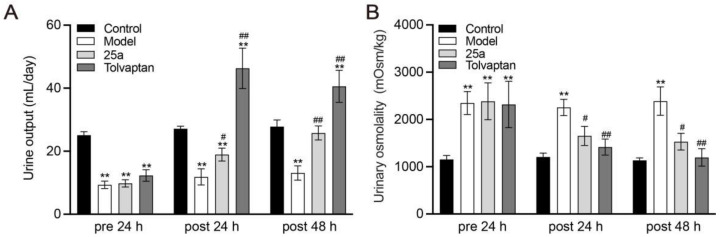
Daily urine output and urinary osmolality in rats. (**A**) Daily urine output. (**B**) Urinary osmolality. Data are presented as means ± SEM, n = 7. ** *p* < 0.01 vs. control group; # *p* < 0.05 and ## *p* < 0.01 vs. model group by one-way ANOVA with Tukey’s HSD post hoc test.

**Figure 3 biomedicines-11-00607-f003:**
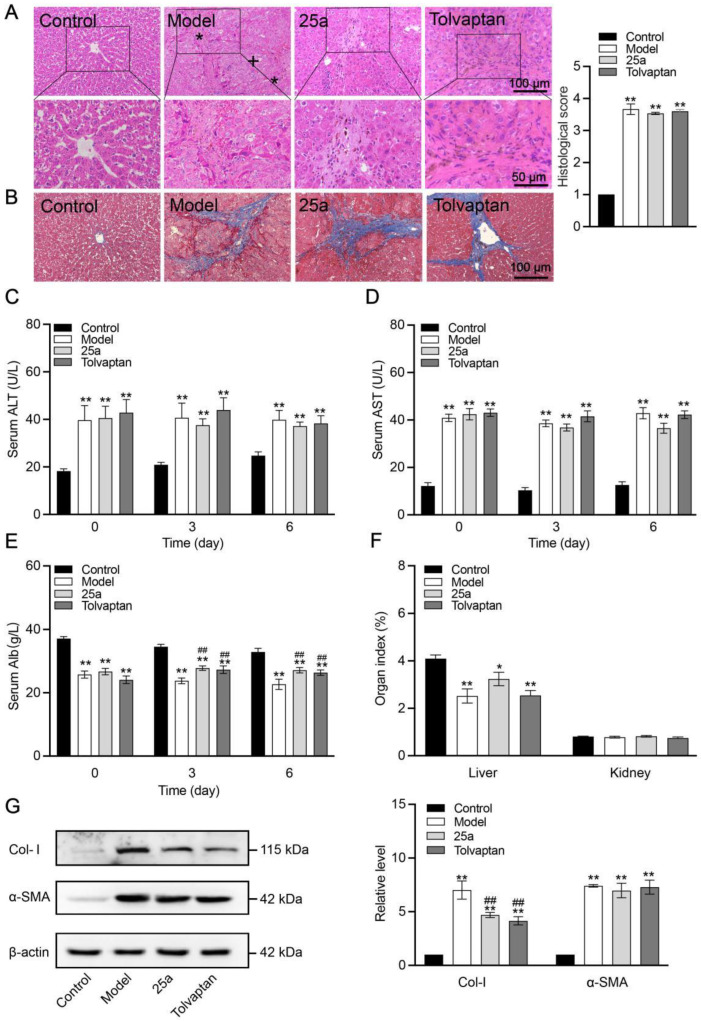
Liver tissue morphology and function in rats. (**A**) H&E-stained liver tissue sections (*, fibrosis; +, hydropic degeneration of hepatocytes) and histological score. (**B**) Masson-stained liver tissue sections. (**C**) Serum ALT. (**D**) Serum AST. (**E**) Serum albumin concentration. (**F**) Organ index. Above data are presented as means ± SE, n = 7. (**G**) Representative Western blots (left) and relative expression level of Col-I and α-SMA in liver quantified from Western blots (right). Data are presented as means ± SEM, n = 3. * *p* < 0.05 and ** *p* < 0.01 vs. control group; ## *p* < 0.01 vs. model group. The data were analyzed by one-way ANOVA with Tukey’s HSD post hoc test.

**Figure 4 biomedicines-11-00607-f004:**
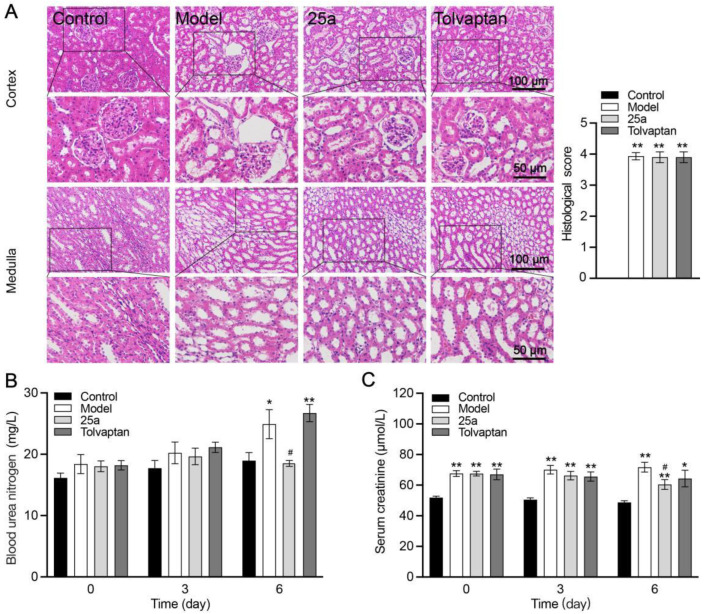
Kidney tissue morphology and function in rats. (**A**) H&E-stained kidney tissue sections and histological scores. (**B**) Blood urea nitrogen (BUN). (**C**) Serum creatinine. Data are presented as means ± SEM, n = 7. * *p* < 0.05 and ** *p* < 0.01 vs. control group; # *p* < 0.05 vs. model group. The data were analyzed by one-way ANOVA with Tukey’s HSD post hoc test.

**Table 1 biomedicines-11-00607-t001:** Serum osmolality and electrolyte concentrations in rats.

	Serum Osmolality (mOsm/kg)	Serum Sodium (mmol/L)	Serum Potassium (mmol/L)	Serum Chloride (mmol/L)
Control	299 ± 2.16	143.28 ± 1.38	5.65 ± 0.24	97.85 ± 1.06
Model	294.33 ± 1.74 *	136.5 ± 4.50 *	4.26 ± 0.79 *	102.4 ± 2.25 **
25a	299.8 ± 0.77 #	144 ± 1.67 ##	5.05 ± 0.31 #	102.28 ± 1.11 **
Tolvaptan	294.85 ± 1.57	145 ± 2.64 #	4.04 ± 0.41	102.75 ± 0.63 **

Data are presented as means ± SEM, n = 7. * *p* < 0.05 and ** *p* < 0.01 vs. control group; # *p* < 0.05 and ## *p* < 0.05 vs. model group by one-way ANOVA with Tukey’s HSD post hoc test.

## Data Availability

Data are contained within the article.

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
