# Peer review of "Urea Transporter Inhibitor 25a Reduces Ascites in Cirrhotic Rats"

_biomedicines, 2023, doi:10.3390/biomedicines11020607_

Round 1
Reviewer 1 Report
Ying et al. studied the effects of 25a, a urea transporter inhibitor, on dimethylnitrosamine (DMN)-induced cirrhosis in rats with water intake limitation. 25a did not change liver function and histology but significantly decreased the protein expressions of type I collagen. 25a reduced ascites by causing diuresis but without electrolyte disturbances.
The main problem with the manuscript is that the statistical analysis in not acceptable. The statistical comparisons should be repeated using appropriate statistical tests.
Another uncertainty is that 25a restored BUN in cirrhotic rats, i.e. likely increased urea/ammonia excretion which might cause negative nitrogen balance. This is a critical issue if the aim is to treat patients with 25a on the long term, which seems to be neglected in the manuscript and also in previous articles of the authors. The effects of 25a on urea/ammonia excretion should be measured.
There are a number of uncertainties concerning the methods and the results which need to be addressed.
Specific comments and suggestions
Why water intake limitation was necessary? Tolvaptan is effective without water intake limitation (PMID: 28751810).
The time-line of the study is not clear. Figures 1-2 seem to present results when treatments with 25a and tolvaptan was started. Other results refer to day 6 after starting the treatments. Please make it clear!
How did you collect blood samples?
Statistical analysis is not acceptable. Two-tailed Student’s t test does not correct for multiple comparison so is can result in a number of false positive results. Statistical analysis should be performed using two-way ANOVA for repeated measurements and an appropriate post hoc test such as Bonferroni's multiple comparisons test.
Please, do not repeat description of methods in the Results section “Rat cirrhotic model was established with intraperitoneal injection of DMN”, etc.
Tolvaptam caused a much greater increase in urine output than 25a but 25a decreased abdominal circumference more than tolvaptam, and the two compounds similarly decreased body weight. These results are contradictory. Tolvaptan increased urine output by about 40 mL (i.e. 40 g) but the decrease in body weight was about 20 g. What is the explanation for this big difference? These results should be discussed in detail.
Treatment with 25a for 6 days may not be sufficiently long enough to demonstrate protective effects on the liver. Since longer treatments are planned, this conclusion seems to be tentative.
Histological sections should be scored and analysed statistically.
What is liver index? The scientific literature use fatty liver index, which is likely different.
The discussion is long and mainly repeats the results. Tolvaptam and 25a had similar effects. What would be the benefit to treat patients with 25a instead of tolvaptam? Please discuss. How does 25a alter urea/ammonia excretion and nitrogen balance? Please discuss.
Author Response
- The main problem with the manuscript is that the statistical analysis in not acceptable. The statistical comparisons should be repeated using appropriate statistical tests.
Response: Thank you for the important suggestion. We reperformed the statistical analysis using One‐way analysis of variance (ANOVA) with Tukey’s HSD post hoc correction, which data are shown in the revised manuscript.
- Another uncertainty is that 25a restored BUN in cirrhotic rats i.e. likely increased urea/ammonia excretion which might cause negative nitrogen balance. This is a critical issue if the aim is to treat patients with 25a on the long term, which seems to be neglected in the manuscript and also in previous articles of the authors. The effects of 25a on urea/ammonia excretion should be measured.
Response: This is a good comment. The reason that we did not measure urea excretion in this work based on our previous studies, which showed that knockout of urea transporters UT-A (Ref 17) and UT-B (Ref 18) did not affect the urea excretion. Functional inhibition of UTs by specific inhibitors did not change urea excretion either (Ref 21). Basing on these data, we assumed the 25a would not affect the urea excretion. Easy evaporation of ammonia from urine samples makes difficulty for exactly calculating ammonia excretion. Our experimental results showed that knockout and functional inhibition of UTs did not increase non-urea solutes excretion, which indicates that 25a might not increase ammonia excretion. However, if you think that it is necessary to measure the effect of 25a on urea/ammonia excretion, we will perform these experiments as soon as possible.
- Why water intake limitation was necessary? Tolvaptan is effective without water intake limitation (PMID: 28751810).
Response: Following the description in the reference (PMID: 23413814), we designed the experimental plan that water was limited to 10 mL plus the amount that the rats took on the previous day. In the presence of cirrhotic ascites, the urine output of rats will be significantly reduced, and water intake needs to be restricted to avoid excessive water intake, which can lead to dilutional hyponatremia and further aggravate the condition.
- The time-line of the study is not clear. Figures 1-2 seem to present results when treatments with 25a and tolvaptan was started. Other results refer to day 6 after starting the treatments. Please make it clear!
Response: In this study, the treatment period was 6 days. Since the diuretic effect of 25a and tolvaptan reached a plateau after 2 days of treatment, we present the results on day 1 before, day 1 after, and day 2 after starting treatment in Figures 1-2. In addition, considering that daily blood sampling affects the general status of rats, we collected venous blood in cirrhotic rats on day 1 before, day 3, and day 6 after starting treatment, and evaluated the electrolyte level, hepatic function and renal function at the same time.
- How did you collect blood samples?
Response: Blood samples were collected from rats by orbital venous plexus puncture with anticoagulation of sodium heparin.
- Statistical analysis is not acceptable. Two-tailed Student’s t test does not correct for multiple comparison so is can result in a number of false positive results. Statistical analysis should be performed using two-way ANOVA for repeated measurements and an appropriate post hoc test such as Bonferroni's multiple comparisons test.
Response: The statistical analysis was reperformed using one‐way analysis of variance (ANOVA) with Tukey’s HSD post hoc correction following the suggestion.
- Please, do not repeat description of methods in the Results section “Rat cirrhotic model was established with intraperitoneal injection of DMN”, etc.
Response: The repetitive description had been removed from the Results section.
- Tolvaptan caused a much greater increase in urine output than 25a but 25a decreased abdominal circumference more than tolvaptan, and the two compounds similarly decreased body weight. These results are contradictory. Tolvaptan increased urine output by about 40 mL (i.e. 40 g) but the decrease in body weight was about 20 g. What is the explanation for this big difference? These results should be discussed in detail.
Response: During the treatment, we did find that tolvaptan caused a much greater increase in urine output than 25a, but 25a decreased abdominal circumference more than tolvaptan, and the two compounds similarly decreased body weight. Our further observation gave us the answer that rats with tolvaptan treatment drank more water basing on the water limitation rule in this study, which is also the reason that tolvaptan increased urine output by about 40 mL, but the decrease in body weight was about 20 g. These results indicates that tolvaptan caused thirst and drinking more water in rats.
- Treatment with 25a for 6 days may not be sufficiently long enough to demonstrate protective effects on the liver. Since longer treatments are planned, this conclusion seems to be tentative.
Response: We agree this comment. In this study, we did not find that 25a had protective effect on the liver in cirrhotic rats. This study is a proof of concept that urea transporter inhibitor reduces ascites in cirrhotic rats. Longer treatment for effect on liver will be performed in the formal preclinical studies.
- Histological sections should be scored and analysed statistically.
Response: Following your suggestion, we have scored and statistically analyzed the histological sections and the added data in the revised manuscript.
- What is liver index? The scientific literature use fatty liver index, which is likely different.
Response: We are sorry for the mistake. “liver index” has been corrected as “liver-to-body-weight ratio”.
- The discussion is long and mainly repeats the results. Tolvaptan and 25a had similar effects. What would be the benefit to treat patients with 25a instead of tolvaptan? Please discuss. How does 25a alter urea/ammonia excretion and nitrogen balance? Please discuss.
Response: We agree with the comment. The discussion has been revised, and the repetitive description had been removed. The benefits of 25a over tolvaptan for treated patients have been well described in the Discussion section of the revised manuscript.
Reviewer 2 Report
In this manuscript the authors have investigated the beneficial effects of the urea transporter inhibitor N-(4-acetamidophenyl)-5-acetylfuran-2-carboxamidein, 25a, in a rat model of cirrhosis with ascites. They show results suggesting a therapeutic effect of 25a by ascites reduction via diuresis and without electrolyte imbalance in cirrhotic rats. No effects are demonstrated in liver cirrhosis although they claim a slower progression of fibrosis because a reduction in protein type 1 collagen. The study is straightforward, and although the aim is important, there are a few points that need attention.The work follows the current search for better diuretics, necessary mostly for patients with refractory ascites where the nowadays used drugs have not response or quick relapses.
Comments
Major:
- Can the authors explain why the animals are fed in pathogen-free conditions.
- How the authors determine when an animal have ascites to start with the treatment?
- The treatment with 25a, tolvaptan or vehicle is not clear. One has the impression that the drugs are administered once and then, measures are taken 24h or 48h after. But later in the manuscript the authors give results taken on day 6 of continuous treatment. Please explain better the treatment schedules/frequency.
- There is no information about the number of animals used in each group at the beguining and at the end of the study. Also, it is important to know how many rats died during DMN model or 25a treatment? Safety/toxicity of the drug is also important and mortality during the experimental procedure is a good parameter.
- How long did the animals stay in metabolic cages? This is a source of stress for them that ould change the results if the animals stay too long.
- One‐way analysis of variance (ANOVA) with Tukey’s HSD post hoc correction is more indicated for the statistical analysis when you are dealing with three or more groups. A sounder result will be obtained if the authors performed the analysis this way.
-The reduction in body weight can be also because of drug toxicity. Please comment
. Did the authors measured the volume of ascites in each group at the end of the experimental process, before getting the liver and kidney samples? This is a more accurate way than to measure changes in abdominal circumference.
- Do the authors have a hypothesis to explain the observed increase in blood albumin level and the diminished liver index in the 25a treated cirrhotic rats?
- I believe the statement of a possible slow-down of cirrhosis progression based only on the reduction on type I collagen measured by Western blot is excessive. Also, some mechanistic explanation of the effect of 25a on reduction of type collagen would be interesting.
- For the readers, the authors could explain briefly what the outstanding pharmacological characteristics are, both in vitro and in vivo of 25a.
- In the discussion, one expects some insights on why this UT inhibitor is having better results than others. A brief review on the ones already under study will be acknowledged.
Minor:
Page 3, 2nd line: the abbreviation of alanine amino transferase is wrong (ALT)
Page 4, Title of section 3.4: and should be but “25a does not reverse cirrhosis and but may slow cirrhotic progression”
Page 7, line 7: Furthermore, should be However, “Furthermore However, 25a decreased BUN and Crea levels on day 6 with continuous treatment.
Page 8, 6th paragraph: “inhibitor” is missing in the sentence: “In conclusion, we discovered the therapeutic effect of urea transporter inhibitor 25a on ascites”
Author Response
Comments
Major:
- Can the authors explain why the animals are fed in pathogen-free conditions.
Response: The reason that animals are fed in pathogen-free conditions is to guarantee no effect of unrelated factors on experimental data.
- How the authors determine when an animal have ascites to start with the treatment?
Response: This is a good question. To make sure that the animals had similar amounts of ascites and treatment was started in similar condition, the rats were weighed and their abdominal circumference was measured daily. The abdominal circumference and body weight were regarded as the index for determining when an animal has ascites and when treatment starts.
- The treatment with 25a, tolvaptan or vehicle is not clear. One has the impression that the drugs are administered once and then, measures are taken 24h or 48h after. But later in the manuscript the authors give results taken on day 6 of continuous treatment. Please explain better the treatment schedules/frequency.
Response: In this study, the treatment period was 6 days. Since the diuretic effect of 25a and tolvaptan reached a plateau after 2 days of treatment, we present the results on day 1 before, day 1 after and day 2 after starting treatment in Figures 1-2. In addition, considering that daily blood sampling affects the general status of rats, venous blood was sampled and tested in cirrhotic rats on day 1 before, day 3, and day 6 after starting treatment.
- There is no information about the number of animals used in each group at the beginning and at the end of the study. Also, it is important to know how many rats died during DMN model or 25a treatment? Safety/toxicity of the drug is also important and mortality during the experimental procedure is a good parameter.
Response: As well known about the success rate of the rat cirrhosis model, 62 male SD rats were used at beginning of this study, 54 of which were used to establish model and 8 were used as control. 16 rats died during the modeling process. Among the 38 surviving rats, 21 rats with similar abdominal circumference, body weight and general condition were selected and randomly divided into 3 groups: model group (n=7), 25a group (n=7) and tolvaptan group (n=7). During the whole treatment period, no rat died in the 25a and tolvaptan treated groups.
- How long did the animals stay in metabolic cages? This is a source of stress for them that ould change the results if the animals stay too long.
Response: The animals stayed in metabolic cages for 8 days, including 2 days for adaption and 6 days for diuretic activity measurement.
- One‐way analysis of variance (ANOVA) with Tukey’s HSD post hoc correction is more indicated for the statistical analysis when you are dealing with three or more groups. A sounder result will be obtained if the authors performed the analysis this way.
Response: The statistical analysis was reperformed using one‐way analysis of variance (ANOVA) with Tukey’s HSD post hoc correction following the suggestion.
- The reduction in body weight can be also because of drug toxicity. Please comment
Response: We performed a preliminary acute toxicity evaluation in mice. A single dose of 2000 mg/kg of 25a was administered and the general appearance, behavior, diet, response to stimuli, mortality, and body weight of mice were continuously monitored for 14 days for evaluating drug toxicity and safety. There was no death and abnormal signs in both male and female mice during the 14-day observation period. The mice treated with 25a were not different from control mice, which indicates that the body weight loss in the present study was due to reduction of ascites rather than drug toxicity.
- Did the authors measured the volume of ascites in each group at the end of the experimental process, before getting the liver and kidney samples? This is a more accurate way than to measure changes in abdominal circumference.
Response: We agree this comment. Actually, at the end of the experiments, we tried to collect the ascites for measuring the ascites volume. But it was very difficult to collect accurate amount of ascites from the rats in the control group and some rats after drug treatment. This is why for us to use change of body weight and abdominal circumference as surrogate indicators to assess ascites volume.
- Do the authors have a hypothesis to explain the observed increase in blood albumin level and the diminished liver index in the 25a treated cirrhotic rats?
Response: It was found that blood albumin level tended to increase with 25a treatment. We consider that this phenomenon could be due to hemoconcentration caused by marked free-water excretion caused by 25a. The significantly lower liver index in the 25a group compared with the control group was due to liver atrophy caused by cirrhosis. The relatively great liver index in the 25a group compared with the model group may be due to the remission of ascites preventing the exacerbation of cirrhosis with 25a treatment.
- I believe the statement of a possible slow-down of cirrhosis progression based only on the reduction on type I collagen measured by Western blot is excessive. Also, some mechanistic explanation of the effect of 25a on reduction of type collagen would be interesting.
Response: We thank you for the valuable comment. According to your suggestions, the description have been updated in the Results and Discussion sections of the revised manuscript.
- For the readers, the authors could explain briefly what the outstanding pharmacological characteristics are, both in vitro and in vivo of 25a.
Response:Following your suggestion, the pharmacological characteristics of 25a in vivo and in vitro have been explained in the Introduction section of the revised manuscript.
- In the discussion, one expects some insights on why this UT inhibitor is having better results than others. A brief review on the ones already under study will be acknowledged.
Response: This is a good suggestion. The advantages of 25a over other UT inhibitors have been described briefly in the Introduction section.
- Minor:
Page 3, 2nd line: the abbreviation of alanine amino transferase is wrong (ALT)
Page 4, Title of section 3.4: and should be but “25a does not reverse cirrhosis and but may slow cirrhotic progression”
Page 7, line 7: Furthermore, should be However, “Furthermore However, 25a decreased BUN and Crea levels on day 6 with continuous treatment.
Page 8, 6th paragraph: “inhibitor” is missing in the sentence: “In conclusion, we discovered the therapeutic effect of urea transporter inhibitor 25a on ascites”
Response: We are sorry for our careless mistakes. Thank you for your carefully checking. These errors have been corrected.
Reviewer 3 Report
Regarding the manuscript entitled "Urea transporter inhibitor 25a reduces ascites in cirrhotic rats", the authors investigated the novel therapy "specific urea transporter inhibitor, 25a) on liver cirrhosis-associated ascites. The authors found the beneficial effect of 25a on reducing liver cirrhosis complications. The study is merit, well-conducted, and promising. Some points should be addressed to improve the overall quality of the work.
1- The methodology section should be written in detail.
2- Referencing of some methods is missed.
3- it is better if the authors supply the images of rats with ascites and liver shape before and after cirrhosis and treatment.
4- result section should be written in detail (fold change or percent) with some statistical information.
5- Higher magnification power of H&E stain is required
6- Morphometric analysis of histopathological findings is required.
7- Limitations of the study must be considered during the evaluation of this study and their impacts on the conclusion.
Author Response
1-The methodology section should be written in detail.
Response: Following the suggestion, we have added more detail description in the methodology section.
2- Referencing of some methods is missed.
Response: Thanks for your reminder. The related references have been added in the revised version.
3- it is better if the authors supply the images of rats with ascites and liver shape before and after cirrhosis and treatment.
Response: This is a good suggestion. Unfortunately, we did not take photos of rats with ascites and liver shape during experiments.
4- result section should be written in detail (fold change or percent) with some statistical information.
Response: Following your suggestion, some quantified data have been added in the Results section.
5- Higher magnification power of H&E stain is required.
Response: We agree with the comment. H&E stain images at higher magnification have been added in Figure 3 and Figure 4.
6- Morphometric analysis of histopathological findings is required.
Response: This is a valuable comment. We have scored and statistically analyzed the histological sections and the added data in the revised manuscript.
7- Limitations of the study must be considered during the evaluation of this study and their impacts on the conclusion.
Response: Limitations of the study have mentioned in Discussion section.
Round 2
Reviewer 1 Report
Dear Authors,
The manuscript very much improved and you answered all my questions. Thank you!
You have clearly shown previously that urea transporter inhibitors do not increase urea excretion. However, BUN steadily increases in the model group that was fully reversed by 25a but not by tolvaptan on day 6. What is the explanation for this finding if urea excretion is not altered by 25a? Can you please add one or two sentences in the manuscript to explain the mechanism of this effect? I am sorry to stress this point, but those readers of your article, who are also not experts in urea transporter inhibitor pharmacology like myself, may also be interested in the explanation.
Congratulations on your excellent project.
Author Response
You have clearly shown previously that urea transporter inhibitors do not increase urea excretion. However, BUN steadily increases in the model group that was fully reversed by 25a but not by tolvaptan on day 6. What is the explanation for this finding if urea excretion is not altered by 25a? Can you please add one or two sentences in the manuscript to explain the mechanism of this effect? I am sorry to stress this point, but those readers of your article, who are also not experts in urea transporter inhibitor pharmacology like myself, may also be interested in the explanation.
Response: Thank you for the suggestion. The explanation of BUN data has been added in the Discussion section.
Reviewer 2 Report
Author's answers are convincing and new added text helps to clarify the meaning of the study.
As a final suggestion, I will introduce the following information in the material and methods section:
-The treatment period was 6 days
- 21 rats with similar abdominal circumference, body weight and general condition were selected and randomly divided into 3 groups: model group (n=7), 25a group (n=7) and tolvaptan group (n=7).
Author Response
As a final suggestion, I will introduce the following information in the material and methods section:
-The treatment period was 6 days
-21 rats with similar abdominal circumference, body weight and general condition were selected and randomly divided into 3 groups: model group (n=7), 25a group (n=7) and tolvaptan group (n=7).
Response: Following your suggestion, the description of animal grouping and treatment period have been added in the Materials and Methods section.